# Estimating Dengue Transmission Intensity in China Using Catalytic Models Based on Serological Data

**DOI:** 10.3390/tropicalmed8020116

**Published:** 2023-02-14

**Authors:** Ning Li, Haidong Li, Zhengji Chen, Huan Xiong, Zhibo Li, Tao Wei, Wei Liu, Xu-Sheng Zhang

**Affiliations:** 1School of Public Health, Kunming Medical University, Kunming 650500, China; 2Centers for Disease Control and Prevention of Yunnan Province, Kunming 650118, China; 3The Affiliated Hospital of Stomatology, Chongqing Medical University, Chongqing 401147, China; 4Kunming Centers for Disease Control and Prevention, Kunming 650228, China; 5Library, Kunming Medical University, Kunming 650500, China; 6Statistics, Modelling and Economics Department, Data, Analytics & Surveillance, UK Health Security Agency, London NW9 5EQ, UK

**Keywords:** Bayesian inference, catalytic model, dengue, mathematical modelling, serological data, transmission intensity

## Abstract

In recent decades, the global incidence of dengue has risen sharply, with more than 75% of infected people showing mild or no symptoms. Since the year 2000, dengue in China has spread quickly. At this stage, there is an urgent need to fully understand its transmission intensity and spread in China. Serological data provide reliable evidence for symptomatic and recessive infections. Through a literature search, we included 23 studies that collected age-specific serological dengue data released from 1980 to 2021 in China. Fitting four catalytic models to these data, we distinguished the transmission mechanisms by deviation information criterion and estimated force of infection and basic reproduction number (*R*_0_), important parameters for quantifying transmission intensity. We found that transmission intensity varies over age in most of the study populations, and attenuation of antibody protection is identified in some study populations; the *R*_0_ of dengue in China is between 1.04–2.33. Due to the scarceness of the data, the temporal trend cannot be identified, but data shows that transmission intensity weakened from coastal to inland areas and from southern to northern areas in China if assuming it remained temporally steady during the study period. The results should be useful for the effective control of dengue in China.

## 1. Introduction

### 1.1. Current Status of Dengue

Dengue is a mosquito-borne virus infectious disease mainly transmitted through the bites of female *Aedes aegypti* and *Aedes albopictus*. Dengue virus (DENV) has four different serotypes (DENV-1, 2, 3 and 4), and both infants and adults are susceptible. The incubation period is generally 4–10 days, and the infection period is 2–7 days [1]. Dengue is one of the 17 neglected tropical diseases in the Neglected Tropical Diseases (NTD) roadmap [2]. In recent decades, the incidence of dengue has risen dramatically. The World Health Organization (WHO) estimates 100 to 400 million cases of infection each year worldwide, and nearly half of the world’s population is at risk of the infection [3]. Moreover, since most cases of dengue manifest as asymptomatic infections (recessive infections), the actual number may exceed the reported number. According to Bhatt et al. [4], about 390 (95% CI: 284, 528) million dengue infections occur each year, of which only about 96 (95% CI: 67, 136) million express clinical symptoms. Dengue has caused a huge burden of disease globally. The year 2019 saw the highest number of dengue cases reported globally in recent memory. Many countries and regions have been affected, with the first recorded transmission of dengue in Afghanistan. The United States alone has reported 3.1 million cases, with more than 25,000 severe cases, while large numbers of cases have also been reported in Bangladesh (101,000), Malaysia (131,000), the Philippines (420,000) and Vietnam (320,000) in Asia [3].

China is in eastern Asia and is adjacent to some Southeast Asia countries that have high dengue incidence. Its southeast coast is a high-incidence area of dengue, especially in Guangdong province [5]. Dengue cases were reported almost every year, and a large outbreak of 45,217 cases occurred in 2014 in China [6]. In addition, in the northern and inland regions, from 2018 to 2020, there were successive outbreaks of dengue in Hebei province [7], Yunnan province [8], Hunan province [9], Hubei province [10], with a total of more than 300 reported cases. China is not the original area of dengue, and most dengue outbreaks in China were caused by imported cases from abroad. Because recessive dengue infections are not detected by symptom screening, it is very likely that a large number of those infections [4] become a source of infection and lead to a significant increase in the positivity rate [11,12,13,14]. The social and economic burdens caused by dengue are getting heavier. In the specific measures to achieve the three major goals of dengue control, the WHO clearly proposed to conduct research on the transmission kinetics of dengue and develop models to quantify the joint method of vaccine and vector control in transmission [15]. In this paper, we used catalytic models based on serological data to estimate dengue transmission intensity and review the changes in dengue transmission in China over the past years.

### 1.2. Dengue Data and Modelling

In the modeling study of infectious diseases, the important parameters used to characterize transmission intensity are the force of infection (FOI, defined as the instantaneous per capita infection rate at which susceptible individuals acquire infection [16]), and basic reproductive number (*R*_0_, defined as the average number of secondary cases caused by an infectious individual entering a susceptible population [17]). The role of FOI has greater significance than the widely used incidence rate because it can distinguish potential age-related changes in infection rates [16]. When an infectious disease first occurs, a patient must infect at least one individual; that is what keeps the epidemic going. However, not every susceptible person who comes into contact with an infected person will be infected, and the probability is determined by FOI. When the susceptible population within a region has got infected and then acquired immunity or died, the proportion of the susceptible population will decline; but due to the supplement of newborn babies and migrant susceptible people from external populations, the proportion of susceptible people may increase again. To characterize the spread potential of an infectious disease under the constantly changing susceptibility, the effective reproduction number, which is defined as a product of basic reproduction number and susceptibility (i.e., $R_{0}\times \mathrm{susceptibility}$), is an appropriate measure. When a population reaches a steady-state for an infectious disease, its effective reproduction is equal to 1, implying the total number of infections is neither increasing nor decreasing. Therefore, *R*_0_ determines not only the growth rate of the infectious disease but also the proportion (1/*R*_0_) of the steady-state population infected by the disease [18].

In China, we learned through a literature search that there were few studies using serological data for dengue modeling. Due to the lag in the infectious disease surveillance and reporting system and the differences in regional regulatory systems, there is a problem with using case notification data in that the number of reports does not match the reality. The spread of dengue shows a high degree of geographic heterogeneity [19] and even needs to be measured in a very fine spatial range [20]. Moreover, in many situations, dengue cases show mild or asymptomatic [4] or are only diagnosed as another viral infection in the clinic, which may cause dengue to be misclassified or difficult to diagnose, even the highly sensitive disease surveillance systems may also underestimate the incidence of dengue [21,22]. Therefore, the use of dengue case notification data for research may underestimate dengue transmission intensity.

Using serological data to estimate dengue transmission intensity has great advantages because it can detect past symptomatic and asymptomatic infection cases [19] and more accurately reflect transmission intensity. A literature review of dengue studies in China showed that most studies used IgM and IgG ELISA (Enzyme-Linked Immunosorbent Assay) data. Although PRNT (Plaque Reduction Neutralization Test) and PCR (Polymerase Chain Reaction) can identify different serotypes of dengue virus, their difficulty and cost are relatively high, while ELISA has the advantages of low cost and high efficiency. Imai [19] used IgG and IgM ELISA, IE (Inhibition ELISA) and PRNT data to estimate dengue transmission intensity, and they showed that the FOI estimated by the ELISA data was equal to the sum of the FOI estimated from the specific serological data. In this study, we used non-serotype-specific data to estimate FOI and *R*_0_.

## 2. Materials and Methods

### 2.1. Literature Search and Data

We searched multiple literature databases for potentially available studies related to dengue serology in China. Since we mainly studied the spread of dengue in China in recent years, articles published before 1980 and articles whose study areas are not in China were excluded. Since a wider age group may not accurately reflect the difference in the seropositivity rate of the age group [19], the studies that had at least five age groups were included. Based on these selection criteria (Figure 1), 23 studies [23,24,25,26,27,28,29,30,31,32,33,34,35,36,37,38,39,40,41,42,43,44,45] were finally included (Appendix A).

The 23 studies involved eight provinces and one region in China, namely Guangdong, Guangxi, Zhejiang, Hunan, Guizhou, Hainan, Yunnan, Taiwan and Hong Kong, with a total of 18 study regions and 31 data sets (Table 1). They are all located south of the Yangtze River in China. Among them, Guangdong, Guangxi, Zhejiang, Hainan, Taiwan and Hong Kong are all coastal provinces, while Hunan, Guizhou and Yunnan are inland provinces. In addition, Yunnan Province and Guangxi Province are adjacent to Southeast Asian countries. Most of the eight provinces or regions are located between 20° and 30° north latitude, with tropical or subtropical seasonal characteristics, high temperature and humid climate.

Those studies reported the survey data on the age-stratified non-serotype-specific prevalence of dengue from 1980 to 2019. In the last column of Table 1, we use “Herd” to indicate that its sample is from the healthy general population; Use “Hospital/CDC” to indicate that the data was collected at the hospital and/or Centers for Disease Control and Prevention (CDC); “Blood donation center” means that the sample population is healthy blood donors in the blood donation center. “Health Checkup Center” means the sample population is the health checkup population at the port health checkup center. In addition, single-year cross-sectional data from 2011 to 2013 can be extracted from the study [27] and from 2013 to 2015 from the study [45]. The study [43] was based on the phased data collected at different stages of a dengue outbreak in the study area. For the three studies, we fitted our models to the data of each year.

### 2.2. Dengue Models

Due to the presence of dengue immune antibodies, the seroprevalence rate of the population increases with age. This rate of change with age can be interpreted as a measure of the “strength” of the spread of dengue in the past. The significant change in the seroprevalence rate of each age group may be further due to a unique change in the risk of infection in a certain age window or caused by a change in unique risk factors that are not related to age, or a combination of the two [46]. Therefore, the seroprevalence rate provides information about the overall cumulative risk of infection experienced by the entire age group [47]. In addition, the individual’s dengue immune antibody level may also decrease with age (antibody protection decay effect). Here, we consider the impact of different infection mechanisms (Model A–D) on the dengue transmission intensity. For Model A, we assume that the FOI does not change with age; that is, the FOI is a constant. For Model B, since the seroprevalence rate of some data sets seems to decrease with age, we assume that antibody protection decays at a constant rate. For Models C and D, we consider that the seropositivity rate changed significantly at a certain age due to changes in the exposure levels or other reasons and introduced the concept of threshold age (*A_crit_*). In view of that changes in population structure can greatly complicate mathematical models and require a large amount of longitudinal population data, we ignore demographic changes such as population mobility and natural birth/death rates. In the following, we give the details of the four models.

#### 2.2.1. Model A: Constant Force of Infection

According to Muench’s catalytic model [48], people in age group *i* changes from a seronegative group to a seropositive group after infection at a rate *λ*, as shown in Figure 2. Here *λ* is used to denote FOI, and the proportions of seronegative and seropositive in the age group *i* are *x(a_i_)* and *z(a_i_)*, respectively.

Since the data is not serotype-specific, we assume that the total FOI of the four serotypes is constant and that individuals receive lifelong immunity after infection. Assuming that *λ* is a constant, the proportion of the seropositive population in age group *i*, *z(a_i_)* is given by:(1)$$z(a_{i})=1-\exp(-\lambda a_{i})$$
Here *a_i_* is the median age of the age group *i*.

#### 2.2.2. Model B: Antibody Protection Decay

There are four serotypes of the dengue virus. A person who is infected with one serotype will have acquired immunity. Although there is serotype cross-immunity, the duration of cross-immunity is very different from person to person [49,50,51]. Studies have shown that in the first six months after primary infection with dengue, the neutralizing antibody (NAb) titers against all serotypes are the highest [52], but after that, NAb titers gradually weaken, which mainly depends on the intensity of dengue and the degree of exposure [53,54,55,56]. According to the data we have obtained, the seroprevalence rate of some data sets decreases with age, which means that there may be a phenomenon that antibody levels decrease with age. Following [19], it is assumed that the immunity level of a seropositive group decays at a rate of *α*, causing individuals to return to the seronegative group (Figure 3).

As shown in Figure 3, the population is divided into the seropositive population (*z(a_i_)*) and seronegative population (*x(a_i_)* = 1 − *z(a_i_*)); the antibody level of the seropositive population decays at a constant rate *α*, while seronegative population may be infected to become seropositive people. By extending Model A, the change in the proportion of the seropositive population in age group *i*, *z(a_i_)* is given by:(2)$$\frac{dz(a_{i})}{da}=\lambda \left[1-z(a_{i})\right]-\alpha z(a_{i})=\lambda -(\lambda +\alpha )z(a_{i})$$

Assuming that both *λ* and *α* are constant, integrating the above formula gives,
(3)$$z(a_{i})=\left\{1-\exp\left[-a_{i}(\lambda +\alpha )\right]\right\}\left(\frac{\lambda }{\lambda +\alpha }\right)$$

#### 2.2.3. Models That Include Threshold Age

For people of different ages (such as the young and old), their risk of getting dengue may be different due to differences in the immune system, lifestyle, and other factors, even if they are exposed to the same environmental conditions. Assuming that the population is homogeneously mixed, for example, young people may reach a wider range of people, while the range of activities of the elderly is limited due to reasons such as study and work. Their potential exposure patterns are also different. In view of these potential factors, we assume that the seropositivity rate may change significantly in a certain age window due to changes in the exposure level or other reasons. We assume that there is a critical age (*A_crit_*) by which the population is divided into two different age groups. Within each age group, the FOI is still assumed to be constant but varies between age groups. Based on this, Model A and Model B are extended to Model C and Model D as follows.

Model C: constant FOI is broken by a critical age
(4)$$z(a_{i})=1-\exp(-\lambda_{1}a_{i}),\;\mathrm{when}\;a\leq A_{crit}\;z(a_{i})=1-\exp(-\lambda_{2}a_{i}),\;\mathrm{when}\;a>A_{crit}$$

Model D: antibody protection decay with constant FOI broken by a critical age
(5)$$z(a_{i})=\left\{1-\exp\left[-a_{i}(\lambda_{1}+\alpha )\right]\right\}\left(\frac{\lambda_{1}}{\lambda_{1}+\alpha }\right),\;\mathrm{when}\;a\leq A_{crit}\;z(a_{i})=\left\{1-\exp\left[-a_{i}(\lambda_{2}+\alpha )\right]\right\}\left(\frac{\lambda_{2}}{\lambda_{2}+\alpha }\right),\;\mathrm{when}\;a>A_{crit}$$

Here, *λ*_1_ and *λ*_2_ are the respective FOI when age *a* is less than or equal to *A_crit_* or greater than *A_crit_*, and the decay rate of antibody protection (*α*) remains constant throughout all the age groups.

### 2.3. Inference Method

To estimate FOI and *R*_0_, we fit the predicted seroprevalence rates of each age group to observed data, wherein the observed proportions of seropositivity in each age group are calculated based on the age-stratified seroprevalence survey data. In this study, we use Bayesian inference to estimate the model parameters [57]. The prior information on parameters is obtained through literature review and experience, and serological data are extracted from the literature as described in Section 2.1 (Appendix A). Combining these with the likelihood function (see Section 2.4), the Markov Chain Monte Carlo (MCMC) method via normal random walk Metropolis-Hastings sampling method is used to generate the posterior distribution of model parameters [57], from which the median and its 95% credible interval (CrI) are obtained. The R statistical software (version 4.2.2 [58]) is used for calculations.

### 2.4. Negative Log-Likelihood (-LnL)

In Bayesian statistics, the likelihood function is calculated through the sample information (observation data). We assume that the probability of seropositive individuals in the age group *i* obeys the Beta Binomial Distribution:$$X_{i}\sim BetaBinomial(N_{i},p_{i},\gamma )$$

Here, *N_i_* is the total number of individuals in age group *i*, *p_i_* is the proportion of seropositive individuals observed in age group *i*, and *γ* represents the overdispersion parameter of the beta-binomial distribution.

Following Imai et al. [19], the total negative log-likelihood function for all age groups, −*LnL*(*p*), is given by:(6)$$-LnL(p)=-\left\{\begin{array}{l} \sum_{i}\log\left\{B\left[X_{i}+m_{i}(\frac{1}{\gamma }-1),N_{i}-X_{i}+(1-m_{i})(\frac{1}{\gamma }-1)\right]\right\}\\ -\log\left\{B\left[m_{i}(\frac{1}{\gamma }-1),(1-m_{i})(\frac{1}{\gamma }-1)\right]\right\} \end{array}\right\}$$
where *B*[*a*,*b*] is the beta function with standard parameters *a* (predicted seropositive number) and *b* (predicted seronegative number), *X_i_* is the number of seropositive individuals in age group *i*, *m_i_* is the predicted proportion of seropositive patients in the age group *i*.

### 2.5. Estimation of Basic Reproduction Number (R_0_)

Assuming that FOI is constant for a certain amount of time, *R*_0_ can be estimated from the formula [46]:(7)$$R_{0}=\frac{1}{1-\int_{0}^{\infty }f(a)z(a)da}\approx \frac{1}{1-\sum_{i=1}^{na}f(a_{i})z(a_{i})}$$

Here *f(a)* is the probability density function of the population age distribution, and *z(a)* is the predicted seroprevalence from the proposed models. We collected the age distribution data of the population in each study area from the National Bureau of Statistics [59] of China and the website [60] of the Red and Black Population Database to calculate *f(a)*. In actual calculations, the age was divided into *n_a_* groups, and Formula (7) sums over these age groups. For the age group *i*, *a_i_* is its median age, and *f*(*a_i_*) is approximated by *f*(*a*_i_) = *n_i_*/*Total_N*, with *n_i_* being the size in age group *i* and *Total_N* being the total number of the population.

### 2.6. Deviation Information Criterion (DIC) and Model Selection

It should be borne in mind that the infectious disease system can be modeled because epidemics involve relatively simple processes that occur within a large number of individuals [18]. In the modeling studies of infectious diseases, models are used to simplify the complex real world, and the performance of model fitting varies among models. To compare model performance, we use the deviation information criterion (DIC) proposed by Spiegelhalter et al. [61], which combines the fit and complexity of the model and can compare models of any structure. The model that has the smallest DIC is the best and will be chosen [57].

Burnham and Anderson [62] suggested models receiving the Akaike Information Criterion (AIC) within 1–2 of the “best” deserve consideration, and 3–7 have considerably less support. According to Spiegelhalter et al. [61], these rules of thumb appear to work reasonably well for DIC. Therefore, in this study, we chose the critical value 3 as the criterion for DIC to select the best model. In other words, when comparing models, we believe that when the DIC difference between different models exceeds 3, there will be a performance difference. Considering that a simple model is more beneficial for result interpretation, for the data set whose DIC difference between the two models is less than 3, the model with a relatively simple structure is selected as the best model.

## 3. Results

Through the literature review, we selected 23 studies from 515 articles, and the study areas included 8 provinces, 14 cities or regions in China. Those studies reported the survey data of dengue age-stratified serological prevalence from 1980 to 2019. For each data set, we estimated its FOI and *R*_0_ using the four models (the details are given in Appendix A). The model fitting curves of the four models to 31 data sets are illustrated in Appendix A. Figure 4 and Appendix A show the comparison results of DIC among the four models, and the final estimates (Table 2) are based on the best models that have the smallest values of DIC.

### 3.1. Model Comparison and Selection

Figure 4 demonstrates the relative DIC values to the best model for the 31 data sets. The results show that the best model for data sets [23,24,25,26,28,29,32,33,36,37,38,39,40,41,42,44,45] is Model C. In addition, Model C is also applicable to the data sets of 2012 and 2013 in the Guangzhou study [27] and the data sets from August to November 2015 and from August to September 2017 in the Kaohsiung study [43]. The best model for 25 data sets among the 31 data sets is Model C: constant FOI is broken by a critical age. This indicates the age effect on the intensity of dengue transmission among these study areas. Model D (Constant FOI broken by a threshold age with antibody protection decay) was applied to the data sets of 2011–2013 in the study of Guangzhou [27], Chenzhou in 2005 [34], and Kaohsiung in 2016 from February to May [43]. Model A (constant FOI without antibody protection decay) applies to data sets from September 2016 to January 2017 in the study in Kaohsiung [43] and the 1980 study in Beihai City, Guangxi [30]. Only the study [35] in Guiyang in 2004–05 is applied to Model B, which assumes constant FOI and antibody protection decay of dengue infection. Overall, model fittings illustrate that age has a universal effect on the transmission of dengue fever, but the protective attenuation effect of antibodies was not absent in the transmission of dengue fever.

### 3.2. Estimates of FOI and R_0_

The estimates of the critical age (*A_crit_*), FOI and *R*_0_ from the best models selected for each data set are shown in Table 2 and Figure 5. Among the 28 data sets that have model C or D as their best models, they can be divided into two groups: For the 19 data sets that have estimated critical ages older than 60 years, the FOI for ages younger than the critical age (*λ*_1_) is weaker than the FOI for ages older than the critical age (*λ*_2_); while for the 9 data sets that have *A_crit_* < 60 years, *λ*_1_ is greater than *λ*_2_. Assuming that the spread of dengue in the study area is in a local potential endemic state, the estimates of *R*_0_ would be greater than 1 (Figure 5). The estimate of *R*_0_ obtained by the best model fittings of 31 data sets was between 1.04 and 2.33 (Table 2). The study [23] conducted in Guangzhou in 1981 had the largest estimate of *R*_0_ = 2.33 (95% CrI: 1.64, 3.50), and the study conducted by Zhou et al. [31] and Gao et al. [34] had the smallest estimate of *R*_0_ = 1.04 (95% CrI: 1.02, 1.10) and *R*_0_ = 1.04 (95% CrI: 1.02, 1.25), respectively. The graph of estimates of *R*_0_ versus the study year from 1980 to 2019 (see Appendix A) showed that *R*_0_ was over 1.5 in 1980 and 1981; since then, it dropped and fluctuated below 1.5.

### 3.3. Time-Space Comparison

Among 23 studies selected for this analysis, only three studies provided multiple years of data. We estimated FOI and *R*_0_, respectively, by using their best models in different years (Figure 6). In the study conducted in Guangzhou [27] from 2011 to 2013, the estimate of *R*_0_ in 2012 was the largest (1.22, 95% CrI: 1.15, 1.40). In the study [43] conducted in Kaohsiung from 2015 to 2017, the estimate of *R*_0_ showed an upward trend, with *R*_0_ = 1.33 (95% CrI: 1.18, 1.82) for the data set collected in August-September 2017. In the study [45] conducted in Hong Kong between 2013 and 2015, *R*_0_ was estimated to be the largest in 2013 (1.12, 95% CrI: 1.06, 1.23). Figure 6 shows that there appears to be no significant time change trend in dengue transmission intensity in the three study areas.

The study areas included 8 provinces and 1 region in China with 31 data sets. These nine regions are geographically illustrated with their respective mean estimate of *R*_0_ in Figure 7. The results suggested there may be an underlying spatial pattern of dengue spread in China: the intensity of dengue infection in coastal areas is generally higher than in inland areas, and the more it extends to the north, the lower the infection intensity.

## 4. Discussion

Based on dengue serological age stratification data extracted from 23 studies in China, we used four catalytic models to distinguish the potential transmission mechanisms of dengue and estimated dengue transmission intensity in study areas. We found that dengue transmission intensity varies among different age groups in most of the study populations, and attenuation of antibody protection is identified in some study populations. Furthermore, we found that *R*_0_ of dengue in China was between 1.04–2.33, which agrees with that Imai’s estimate for China (1.15–2.88) [63] and is comparable with that in Singapore (1.24–1.48) and Vietnam (1.76–1.85), but lower than that in Thailand (1.96–3.96) and Brazil (2.07–2.60) [19]. Our estimate of *R*_0_ should provide useful information for the herd immunity threshold level and the effectiveness of vaccination or vector control measures required to control the spread of dengue in China [64].

Our study showed that there was a strong relationship between age and dengue transmission in the population. The model fitting indicated that the dengue transmission intensity changed at a critical age *A_crit_*. Age could be used as a scale to quantify the exposure history in the past time, so we simply introduced *A_crit_* to model the potential change in transmission intensity with age. The performance of Model C and Model D, which include a critical age in transmission intensity, was better in 28 of the 31 data sets from the 23 studies. This suggests that populations in those study areas have generally experienced changes in the risk of dengue transmission over a period of time. The critical age might be the time when the risk of dengue transmission changed in the study population. In addition, the existence and identification of the critical age provide a basis for the optimal formulation of dengue prevention and control measures. For example, for the study [41] conducted in 2019 in Xishuangbanna, Yunnan Province, *A_crit_* was estimated to be 60.5 years (95% CrI: 10.9, 83.3), and the FOI was 0.0086 (95% CrI: 0.0040, 0.1340) and 0.0199 (95% CrI: 0.0031, 0.1439) for ages younger and older than the critical age, respectively; this indicates that the risk of dengue transmission was greater in the population aged older than 60.5 years, and the prevention and control of dengue fever should be focused on the elderly population. On the contrary, for the study [33] conducted in Cixi City, Zhejiang Province, in 2004, *A_crit_* was estimated to be 14 years (95% CrI: 9.7, 81.9), and the FOI was 0.0238 (95% CrI: 0.0016, 0.1317) and 0.0029 (95% CrI: 0.0015, 0.0205) for ages younger and older than 14 years, respectively, suggesting that children and adolescents were at greater risk of dengue transmission, and should be the focus of dengue prevention and control.

Primary infection with one serotype is often able to provide life-long immunity against the reinfection of the same serotype. However, cases of homotypic reinfection confirmed by reverse transcriptase-polymerase chain reaction (RT-PCR) have recently been observed in Nicaragua [65]. Severe DENV-2 transmission has also appeared in the DENV-2 antibody population in Iquitos, Peru [66,67]. The increase in infection in the older age group may be due to a significant change in the risk of dengue infection in the study group 51 years ago, as well as a decrease in antibody levels in the older age group. This indicates that isotype immunity may not be able to obtain complete protection, especially when the specific virulence of the virus strain is high, the infectivity is high, or the quality of host immunity is poor [68]. In addition, because there are four types of DENV, specific immune antibodies acquired through infection with one serotype might provide only partial protection against the other three serotypes, and it is still possible to infect a different serotype virus for a second time in the future. Due to the lack of specific serological data on dengue, it is difficult to estimate the transmission intensity of each serotype in this study. However, the potential relationship between the attenuation of antibody protection and the transmission of dengue fever in the population could be identified through mathematical modeling. Our comparison of model performance based on the DIC value illustrates this: The study [34] in Chenzhou, Zhejiang Province in 2005, provided evidence of this phenomenon. The applicable model for the data set from this study is Model D, with the estimated critical age being 51.7 years (95% CrI: 10.3, 83.7) and the FOI being 0.0047 (95% CrI: 0.0011, 0.0537) and 0.0061 (95% CrI: 0.0006, 0.1033) for ages younger and older than the critical age, respectively; its antibody protection is estimated to decay at a rate of 0.08 (95% CrI: 0.02, 0.10) per year.

Although we only estimated the total FOI of all serotypes in the study areas based on non-serotype-specific data, these data were still sufficient to assess the heterogeneity of overall dengue transmission intensity between different regions and populations. Related studies have demonstrated that FOI estimated from non-serotype-specific data is consistent with the sum of FOI estimated from PRNT data [27]. Although the data sets collected cover the period from 1981 to 2019, these data sets were obtained from different geographical locations across China. Among 23 studies collected, only three studies [27,43,45] provided the serological surveys over multiple years in the same locations; within the short time gaps only over 3 years, the data sets from the three studies cannot show any clear temporal changes in transmission intensity (Figure 6). If seroprevalence survey data over a long period, say more than 5 years were available, the potential periodicity or another temporal trend in dengue transmission intensity could be inferred (e.g., [19]). In view of temporally steady transmission intensity [19], the average transmission intensity over different locations within the same province and different study years (if there were more than one study in one province) showed the geographical patterns: It weakened from coastal areas to inland areas, and from southern areas to northern areas (Figure 7). This might reflect that the large and dense populations, including many foreigners in the southeast coastal areas of China, provided certain beneficial conditions for the spread of dengue. Compared to other inland provinces, Yunnan province had a relatively high transmission intensity. This might reflect its specialty: located in the southwest of China and bordered by Myanmar, Laos and Vietnam, which had a high incidence of dengue [19,20] and frequent population movements, trade, and cultural exchanges. The highest mean estimate of basic reproduction number comes from Guangxi Zhuang Autonomous Region; this could be the consequence of the following factors of the region: bordering with Vietnam and being parts of the coastline so that the pressure for imported cases to spread locally is enormous, and the hot and humid climate that is good for mosquitos to live and grow and therefore is more conducive to the spread of dengue fever.

It was reported that the force of infection has declined while the average age of dengue hemorrhagic fever (DHF) cases in Thailand has increased from 8.1 to 24.3 years over the last four decays. This is mainly driven by the decreased birth and death rate [69,70]. The limited data we collected in this study cannot show any clear temporal change in transmission intensity in China, as shown in Figure 6 and Appendix A. Further, our data are age-stratified serological data and cannot directly be used for analyzing the age of dengue hemorrhagic fever (DHF) cases. With the similar demographic transition due to decreased birth and death rates, it is interesting to investigate whether a similar change pattern in the age of dengue cases also occurred in China. This information is important for the control and prevention of dengue in China and is surely a future topic of investigation.

The advantage of using serological data for inferring the burden and transmission intensity of dengue is that it is not affected by infectious disease surveillance systems and case reporting systems. With more than 75% of people infected with dengue having no clinical symptoms [4], serological data can more accurately estimate the actual number of cases. However, there are still some problems. The main problem lies in the differences in the methods used in the studies included. In the 23 studies, seroprevalence surveys sampled different populations and used serum samples collected for different purposes, which might not be representative of the total population in the study area. For example, the sample populations of the studies [24,25] were entry and exit personnel at Zhuhai Port; the sample populations of studies [28,41] were the blood donors in the blood donation center; the sample populations of studies [39,40] were entry-exit personnel at port health examination centers; the sample population of the study [45] was patients sent to the Prince of Wales Hospital in Hong Kong for diagnosis. The use of convenient serum samples could increase the amount of serological data, but the potential bias introduced by such sampling must be considered when analyzing such data.

There are some other limitations in our research. First, when using the models to estimate *R*_0_ through FOI, we assumed that dengue was in endemic balance, which means that the estimates of *R*_0_ for all data sets were greater than 1. However, this was obviously not applicable to areas where dengue was not endemic or the cases were mainly imported from outside. Secondly, the literature review for dengue showed that there were still few studies in China using serological data as a tool to monitor dengue transmission. Most model studies used case notification data, and its reliability largely depended on the quality of the infectious diseases surveillance system and reporting system [71]. A better understanding of changes in transmission intensity can not only improve estimates of the burden of disease caused by dengue but also help policymakers develop effective prevention and control plans. Therefore, we advocate the more extensive and routine use of serological surveys as a monitoring tool to provide valuable data for the study of infectious disease as dengue.

## Figures and Tables

**Figure 1 tropicalmed-08-00116-f001:**
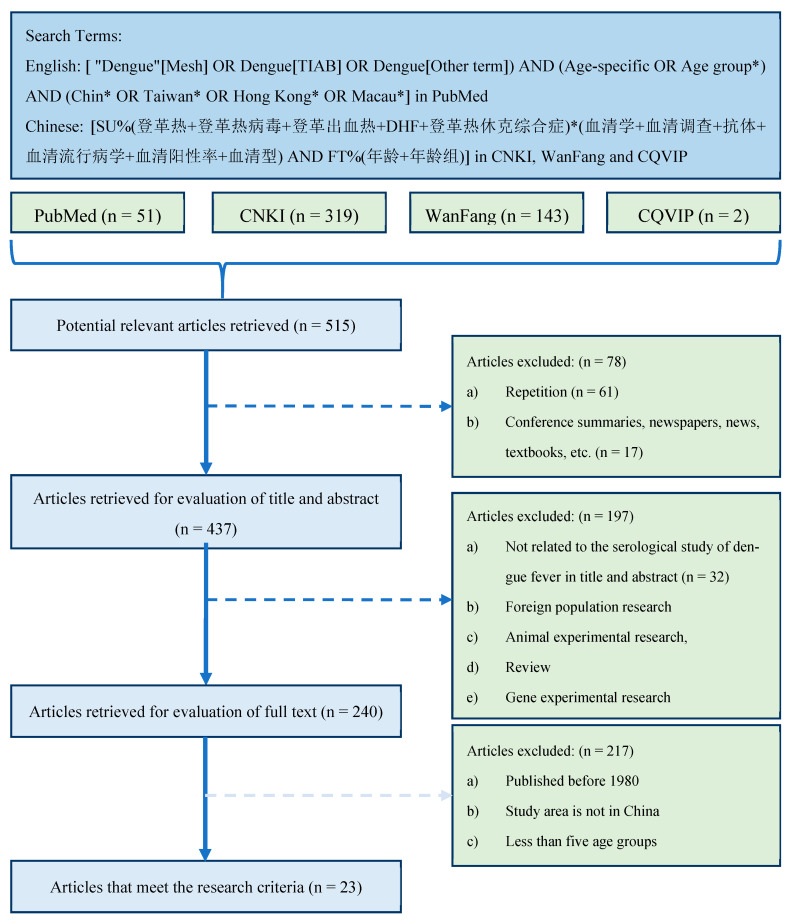
Literature search and screening process. Notice: (1) In the search query [ "Dengue"[Mesh] OR Dengue[TIAB] OR Dengue[Other term]) AND (Age-specific OR Age group*) AND (Chin* OR Taiwan* OR Hong Kong* OR Macau*], *, an asterisk, is the truncation symbol in PubMed, which was used at the end of a word to search for all terms that begin with that basic word root. (2) The search query [SU%(登革热+登革热病毒+登革出血热+DHF+登革热休克综合症)*(血清学+血清调查+抗体+血清流行病学+血清阳性率+血清型) AND FT%(年龄+年龄组)] in CNKI database can be translated into [SU%(Dengue + Dengue virus + Dengue hemorrhagic fever +DHF+ Dengue shock syndrome)*(Serology + serum survey + antibody + seroepidemiology + Seropositive rate + Serotype) AND FT%(age + age group)], where SU stands for subject, or TITLE, ABSTRACT, and/or KEYWORD; % stands for INCLUDE; + stands for the boolean operator OR; * stands for the boolean operator AND; FT stands for FULL TEXT. (3) The search queries for other two Chinese literature databases are similar to the one used in CNKI database.

**Figure 2 tropicalmed-08-00116-f002:**
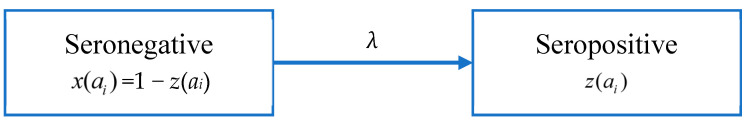
Model A: the catalytic model that assumes constant FOI without antibody decay.

**Figure 3 tropicalmed-08-00116-f003:**
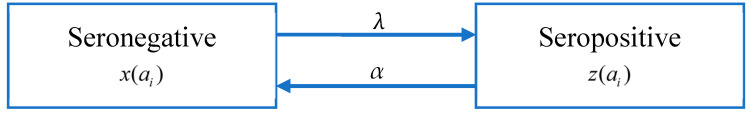
Model B: The catalytic model that assumes constant FOI and antibody decay.

**Figure 4 tropicalmed-08-00116-f004:**
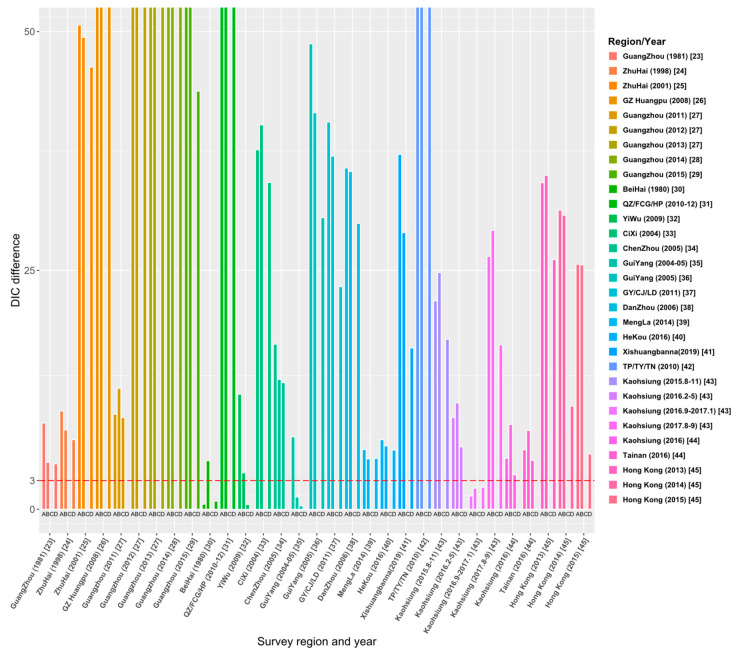
Comparison of DIC values over four models for 31 data sets. The ordinate ‘DIC difference’ is the difference between the DIC and the minimum DIC obtained by fitting the four models to data sets.

**Figure 5 tropicalmed-08-00116-f005:**
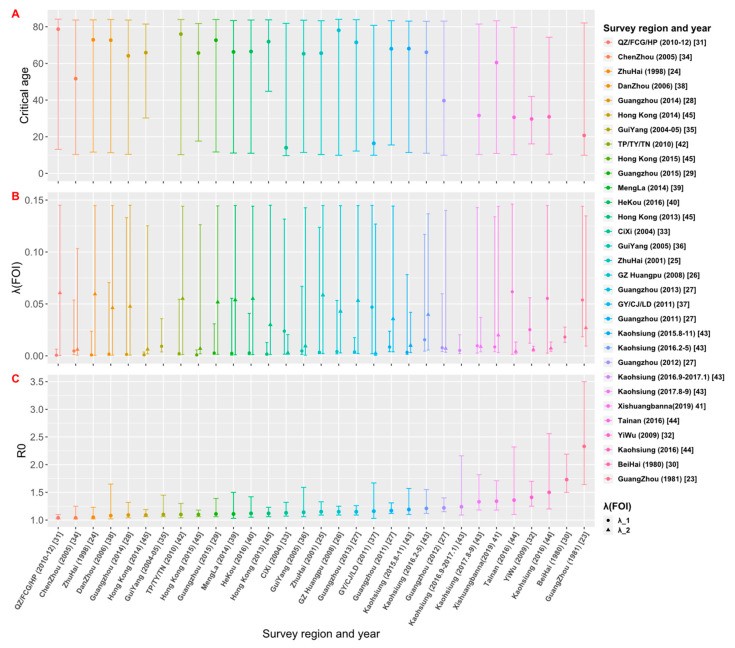
Estimates of (**A**) critical age (*A_crit_*), (**B**) force of infection (FOI), and (**C**) basic reproduction number (*R*_0_) based on the best model fittings to data sets. The estimates from 31 data sets are arranged in the ascending order of *R*_0_.

**Figure 6 tropicalmed-08-00116-f006:**
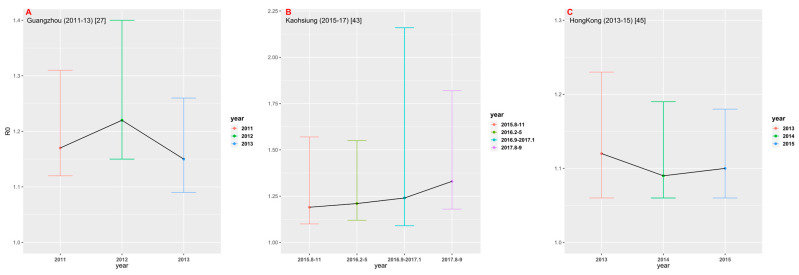
Comparison of *R*_0_ in different time periods in the three regions: (**A**) Guangzhou; (**B**) Kaohsiung; and (**C**) Hong Kong.

**Figure 7 tropicalmed-08-00116-f007:**
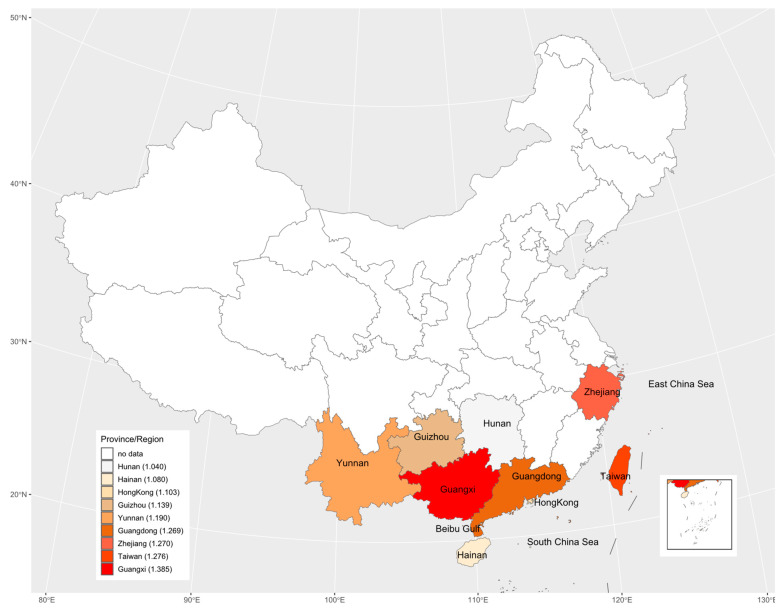
The geographic distribution of the basic reproduction number across the nine study areas. The legend indicates that from top to bottom, the darker the color is, the greater the basic reproduction number is. The numbers in legend brackets are the average of the basic reproduction numbers for each province/region.

**Table 1 tropicalmed-08-00116-t001:** 23 studies for estimation of dengue transmission intensity in China.

Survey Region	Reference	Survey Year	Age Range	No. of Samples	No. of Positives	Testing Method	* Circulating Serotype	Source of Sample Population
Guangdong Province								
Guangzhou	Huang Y et al. [23]	1981	5–50+	174	86	HI	DENV-1,2,3,4	Herd
Zhuhai	Li Z et al. [24]	1998	10–60+	374	4	ELISA	NA	Herd
Zhuhai	Yang Z et al. [25]	2001	10–50+	558	51	ELISA	NA	Herd
^#^ GZ Huangpu	Zheng X et al. [26]	2008	0–71+	324	55	ELISA/RT-PCR	NA	Herd
Guangzhou	Cao Y et al. [27]							
		2011	0–60+	2075	200	ELISA	NA	Hospital/CDC
		2012	0–60+	1201	192	ELISA	NA	Hospital/CDC
		2013	0–60+	1235	124	ELISA	NA	Hospital/CDC
Guangzhou	Li S et al. [28]	2014	18–60	4000	131	ELISA/PCR	DENV-1,2	Blood Donation Center
Guangzhou	Jing Q et al. [29]	2015	0–60+	850	56	ELISA/IFA test	DENV-1,2,3,4	Herd
Guangxi Province								
Beihai	Tian X et al. [30]	1980	0–40	435	116	HI	DENV-2	Sentinel Hospital
^$^ QZ/FCG/HP	Zhou K et al. [31]	2010–2012	0–79	1800	37	ELISA	NA	Herd
Zhejiang Province								
Yiwu	Sun J et al. [32]	2009	0–80+	365	102	ELISA	DENV-3	Herd
Cixi	Cen D et al. [33]	2004	0–80+	520	35	IFA	NA	Herd
Hunan Province								
Chenzhou	Gao L et.al. [34]	2005	0–80+	488	7	ELISA	NA	Herd
Guizhou Province								
Guiyang	Gao R et al. [35]	2004–2005	0–50+	2281	197	ELISA	NA	Herd
Guiyang	Tian H et al. [36]	2005	0–60+	755	55	ELISA	NA	Herd
^&^ GY/CJ/LD	Jiang W et al. [37]	2011	5–60+	530	11	ELISA	NA	Herd
Hainan Province								
Danzhou	Jin Y et al. [38]	2006	0–60+	431	7	ELISA	NA	Herd
Yunnan Province								
Mengla	Lu Y et al. [39]	2014	0–60	182	3	ELISA	NA	Health Checkup Center
Hekou	Pu L et al. [40]	2016	0–60	203	9	ELISA/ RT-PCR	NA	Health Checkup Center
Xishuangbanna	Li L et al. [41]	2019	18–60	2254	484	ELISA	NA	Blood Donation Center
Taiwan Province								
^%^ TP/TY/TN	Lee YH et al. [42]	2010	0–70+	1308	44	ELISA	NA	Herd
Kaohsiung	Tsai JJ et al. [43]							
		2015.8–11	0–89	417	48	ELISA	DENV-1,2	Herd
		2016.2–5	0–89	294	36	ELISA	DENV-1,2	Herd
		2016.9–2017.1	0–59	226	23	ELISA	DENV-1,2	Herd
		2017.8–9	20–89	153	28	ELISA	DENV-1,2	Herd
Kaohsiung and Tainan	Pan YH et al. [44]							
Kaohsiung		2016	40–80+	1498	595	ELISA	DENV-1,2,3,	Herd
Tainan		2016	40–80+	2603	291	ELISA	DENV-1,2,3,4	Herd
Hong Kong	Lee P et al. [45]							
		2013	1–66+	700	24	ELISA	NA	Hospital
		2014	1–66+	700	32	ELISA	NA	Hospital
		2015	1–66+	700	31	ELISA	NA	Hospital

* Circulating serotype is the dengue serotype detected in the study or the main serotype currently circulating, and NA indicates the serotype that was not detected or mentioned in the study. ^#^ GZ Huangpu represents Huangpu District in Guangzhou city; ^$^ QZ/FCG/HP stands for Qinzhou City, Fangchenggang City and Hepu County in Guangxi; ^&^ GY/CJ/LD represents Yunyan District of Guiyang City, Congjiang County of Southeast Guizhou Province and Luodian County of Guizhou Province; ^%^ TP/TY/TN indicates Taipei, Taoyuan and Tainan of Taiwan Province.

**Table 2 tropicalmed-08-00116-t002:** Summary of estimation results from the best model fitting to 31 data sets.

Survey Region	Reference	Survey Year	*A_crit_* (95% *CrI*)	*λ* (95% *CrI*) Year^−1^		*R*_0_ (95% *CrI*)	α (95% *CrI*) Year^−1^	Applicable Model
Guangdong Province				*λ* _1_	*λ* _2_			
Guangzhou	Huang Y et al. [23]	1981	20.7 (9.9, 82.1)	0.0538 (0.0184, 0.1440)	0.0269 (0.0094, 0.1348)	2.33 (1.64, 3.50)	—	Model C
Zhuhai	Li Z et al. [24]	1998	72.9 (11.6, 83.8)	0.0009 (0.0003, 0.0236)	0.0595 (0.0004, 0.1447)	1.05 (1.02, 1.23)	—	Model C
Zhuhai	Yang Z et al. [25]	2001	65.6 (10.3, 83.3)	0.0034 (0.0021, 0.1238)	0.0585 (0.0022, 0.1448)	1.15 (1.09, 1.33)	—	Model C
^#^ GZ Huangpu	Zheng X et al. [26]	2008	78.1 (9.9, 84.1)	0.0041 (0.0019, 0.0533)	0.0428 (0.0028, 0.1446)	1.15 (1.09, 1.25)	—	Model C
Guangzhou	Cao Y et al. [27]	2011–2013						
		2011	68.0 (15.5, 83.3)(15.5, 83.3)	0.0085 (0.0040, 0.0237)	0.0356 (0.0039, 0.1442)	1.17 (1.12, 1.31)	0.04 (0.00, 0.10)	Model D
		2012	39.7 (9.9, 83.1)	0.0078 (0.0041, 0.0598)	0.0071 (0.0034, 0.1400)	1.22 (1.15, 1.40)	—	Model C
		2013	71.5 (12.2, 83.9)	0.0037 (0.0024, 0.0175)	0.0532 (0.0022, 0.1447)	1.15 (1.09, 1.26)	—	Model C
Guangzhou	Li S et al. [28]	2014	64.2 (10.4, 83.7)	0.0014 (0.0007, 0.1331)	0.0476 (0.0007, 0.1450)	1.09 (1.04, 1.32)	—	Model C
Guangzhou	Jing Q et al. [29]	2015	72.7 (11.7, 84.0)	0.0027 (0.0014, 0.0308)	0.0517 (0.0014, 0.1444)	1.11 (1.06, 1.39)	—	Model C
Guangxi Province								
Beihai	Tian X et al. [30]	1980	—	0.0181 (0.0130, 0.0277)		1.73 (1.50, 2.19)	—	Model A
**^$^** QZ/FCG/HP	Zhou K et al. [31]	2010–2012	78.7 (13.1, 84.2)	0.0006 (0.0003, 0.0064)	0.0606 (0.0003, 0.1450)	1.04 (1.02, 1.10)	—	Model C
Zhejiang Province								
Yiwu	Sun J et al. [32]	2009	29.7 (16.1, 42.0)	0.0252 (0.0122, 0.0559)	0.0061 (0.0043, 0.0093)	1.41 (1.25, 1.70)	—	Model C
Cixi	Cen D et al. [33]	2004	14.0 (9.7, 81.9)	0.0238 (0.0016, 0.1317)	0.0029 (0.0015, 0.0205)	1.13 (1.07, 1.32)	—	Model C
Hunan Province								
Chenzhou	Gao L et al. [34]	2005	51.7 (10.3, 83.7)	0.0047 (0.0011, 0.0537)	0.0061 (0.0006, 0.1033)	1.04 (1.02, 1.25)	0.08 (0.02, 0.10)	Model D
Guizhou Province								
Guiyang	Gao R et al. [35]	2004–2005	—	0.0092 (0.0038, 0.0358)		1.10 (1.06, 1.45)	0.08 (0.02, 0.10)	Model B
Guiyang	Tian H et al. [36]	2005	65.3 (11.4, 83.6)	0.0047 (0.0014, 0.0669)	0.0094 (0.0006, 0.1426)	1.14 (1.06, 1.59)	—	Model C
**^&^** GY/CJ/LD	Jiang W et al. [37]	2011	16.4 (9.9, 80.8)	0.0469 (0.0020, 0.1449)	0.0025 (0.0004, 0.1269)	1.16 (1.03, 1.67)	—	Model C
Hainan Province								
Danzhou	Jin Y et al. [38]	2006	72.7 (11.3, 84.0)	0.0017 (0.0004, 0.0706)	0.0462 (0.0006, 0.1447)	1.08 (1.02, 1.65)	—	Model C
Yunnan Province								
Mengla	Lu Y et al. [39]	2014	66.3 (11.1, 83.4)	0.0024 (0.0007, 0.0554)	0.0538 (0.0010, 0.1448)	1.11 (1.03, 1.50)	—	Model C
Hekou	Pu L et al. [40]	2016	66.5 (11.0, 83.7)	0.0028 (0.0011, 0.0409)	0.0551 (0.0012, 0.1441)	1.12 (1.05, 1.42)	—	Model C
Xishuangbanna	Li L et al. [41]	2019	60.5 (10.9, 83.3)	0.0086 (0.0040, 0.1340)	0.0199 (0.0031, 0.1439)	1.34 (1.18, 1.71)	—	Model C
Taiwan Province								
**^%^** TP/TY/TN	Lee YH et al. [42]	2010	76.0 (10.2, 84.0)	0.0021 (0.0008, 0.0545)	0.0551 (0.0012, 0.1441)	1.10 (1.04, 1.30)	—	Model C
Kaohsiung	Tsai JJ et al. [43]	2015–2017						
		2015.8–11	68.1 (11.4, 83.1)	0.0034 (0.0014, 0.0782)	0.0099 (0.0031, 0.0419)	1.19 (1.10, 1.57)	—	Model C
		2016.2–5	66.1 (11.0, 83.0)	0.0155 (0.0046, 0.1169)	0.0397 (0.0056, 0.1368)	1.21 (1.12, 1.55)	0.07 (0.01, 0.10)	Model D
		2016.9–2017.1	—	0.0052 (0.0021, 0.0203)		1.24 (1.09, 2.16)	—	Model A
		2017.8–9	31.6 (10.3, 81.5)	0.0097 (0.0026, 0.1428)	0.0089 (0.0039, 0.0370)	1.33 (1.18, 1.82)	—	Model C
Kaohsiung and Tainan	Pan YH et al. [44]	2016						
Kaohsiung			30.9 (10.5, 74.3)	0.0553 (0.0026, 0.1447)	0.0073 (0.0043, 0.0133)	1.50 (1.20, 2.56)	—	Model C
Tainan			30.6 (10.2, 79.7)	0.0617 (0.0015, 0.1461)	0.0042 (0.0019, 0.0133)	1.36 (1.10, 2.32)	—	Model C
Hong Kong	Lee P et al. [45]	2013–2015						
		2013	71.9 (44.8, 83.8)	0.0017 (0.0008, 0.0128)	0.0299 (0.0013, 0.1450)	1.12 (1.06, 1.23)	—	Model C
		2014	65.9 (30.2, 81.5)	0.0010 (0.0005, 0.0037)	0.0063 (0.0019, 0.1252)	1.09 (1.06, 1.19)	—	Model C
		2015	65.7 (17.6, 81.8)	0.0009 (0.0005 0.0057)	0.0070 (0.0019, 0.1263)	1.10 (1.06, 1.18)	—	Model C

Note: **^#^** GZ Huangpu represents Huangpu District in Guangzhou city; ^$^ QZ/FCG/HP stands for Qinzhou City, Fangchenggang City and Hepu County in Guangxi; **^&^** GY/CJ/LD represents Yunyan District of Guiyang City, Congjiang County of Southeast Guizhou Province and Luodian County of Guizhou Province; ^%^ TP/TY/TN indicates Taipei, Taoyuan and Tainan of Taiwan Province.

## Data Availability

Data from previously published studies are all listed in the references and provided in Appendix A.

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
