# Peer review of "Estimating Dengue Transmission Intensity in China Using Catalytic Models Based on Serological Data"

_tropicalmed, 2023, doi:10.3390/tropicalmed8020116_

Round 1

Reviewer 1 Report

Minor language and grammar corrections are required. Few highlighted. Please go through the entire text ofnthe manuscript and correct those.

Author Response

We have checked through the entire manuscript and corrected the minor language and grammar problems. The details are listed below.

Line 14: “>75%” has been changed to “more than 75%

Lines 38-44: “…the incidence … dramatically globally. … had clinical symptoms” has been changed to

“…the incidence of dengue has risen dramatically. The World Health Organization (WHO) estimates 100 to 400 million cases of infection each year worldwide, and nearly half of the world’s population is at risk of the infection [3]. Moreover, since most cases of dengue manifest as asymptomatic infections (recessive infections), the actual number may exceed the reported number. According to Bhatt et al. [4], about 390 (95%CI: 284, 528) million dengue infections occur each year, of which only about 96 (95%CI: 67, 136) million express clinical symptoms

Lines 58-60: “Because of a large number of … in the positive rate” has been reconstructed as:

“Because recessive dengue infections are not detected by symptom screening, it is very likely for the large number of those infections [4] to become a source of infection and to lead to a significant increase in the positivity rate”

Lines: 78-80: “When the susceptible …will decline” has been changed to

When the susceptible population within a region has got infected and then acquired immunity, or died, the proportion of susceptible population will decline”.

Line 82: ”will increase” has been changed to “may increase”.

Lines 82-84: “To characterize the spread potential … is an appropriate measure.”  has been changed to

“To characterize the spread potential of an infectious disease under the constantly changing susceptibility, the effective reproduction number which is defined as a product of basic reproduction number and susceptibility (i.e.,) is an appropriate measure.”

Line 114: “we mainly study” has been changed to “we mainly studied”.

Line 118: “are included” has been changed to “were included”.

Lines 153: “will increase” has been changed to “increases”.

Line 164: “we considered” has been changed to “we consider”.

Line: 172: “will change” has been changed to “changes”.

Line 193: “immune level” has been changed to “immunity level”; “will decay” to “decays”.

Lines 223-224: “Here, l1 is …than Acrit,” has been changed to “Here, l1 and l2 are the respective FOI when age a is less than or equal to Acrit, or greater than Acrit”.

Line 236: “95% confidence interval (CI)” has been changed to “95% credible interval (CrI)”. In the whole paper, all “95% CI” have been changed to “95% CrI”, because these intervals are obtained through MCMC samplings.

Line 239: “In Bayesian statistics, we can calculate … sample through...” has been changed to “In Bayesian statistics, the likelihood function is calculated through…”.

Line 253: “Estimate the basic reproduction number” has been changed to “Estimation of basic reproduction number”.

Lines 295-297: “In addition, the data sets of … also applicable to Model C” has been changed to “In addition, Model C is also applicable to the data sets ….”.

Lines 297-298: “Among the 31 data sets, the best model … a critical age.” has been changed to “The best model for 25 data sets among the 31 data sets is Model C: constant FOI broken by a critical age.

Line 337: “collected” has been changed to “selected”.

Line 338: “using data fitting optimal models” has been changed to “using their best models”.

Line 346: “include” was changed to “included”.

Reviewer 2 Report

In this manuscript, the authors estimated the transmission intensity of dengue infection in China using catalytic models. Using literature search, they collected seroprevalence data on dengue for nine Chinese provinces and regions for the period from 1980 to 2021. They compared four versions of catalytic models, and estimated both the force of infection and R0.

I have the following comments:

(1)   I think the authors did not consider the difference in immunity for primary versus secondary dengue infection. Primary infection with one serotype A is often able to provide life-long immunity against the reinfection of the same serotype A but not the other three serotypes. Secondary dengue infection is often able to provide full protection against all four serotypes. Could you include this biological factor in your catalytic models?

(2)   The authors analyzed a very long time period covering several decades. It will be interesting if the authors could discuss the influence of demographic changes (e.g., changes in birth and death rates) on dengue transmission intensity in China.

These two references would be useful to compare: (1) Assessing the role of multiple mechanisms increasing the age of dengue cases in Thailand, PNAS (2022); (2) The impact of the demographic transition on dengue in Thailand: Insights from a statistical analysis and mathematical modeling. PLoS Medicine (2009).

(3)   In line 236, do you mean “credible interval”?

(4)   In line 99, dengue case notification data is useful to provide reliable estimation of the dengue force of infection, if the model accounts for the under-reporting ratio.

Author Response

I have the following comments:

(1)   I think the authors did not consider the difference in immunity for primary versus secondary dengue infection. Primary infection with one serotype A is often able to provide life-long immunity against the reinfection of the same serotype A but not the other three serotypes. Secondary dengue infection is often able to provide full protection against all four serotypes. Could you include this biological factor in your catalytic models?

Answer: The relationship between primary and secondary dengue infection was one hot topic of dengue research as secondary infections with dengue virus are more severe than primary infections for which a common explanation focuses on antibody-dependent enhancement [Reich et al 2013]. On lines 393-403, the relevant contents about protection against multiple serotypes of dengue virus have been discussed. In our simple catalytic models the complex interactions between primary and secondary infection and between four serotypes cannot be covered. For this, the detailed transmission dynamic models such as those used by Reich et al 2013 and Huang et al 2022 will be required.  In the revised manuscript, the following was added on line 393:

“Primary infection with one serotype is often able to provide life-long immunity against the reinfection of the same serotype, however, cases of …”.

Reich NG et al. Interactions between serotypes of dengue highlight epidemiological impact of cross-immunity J R Soc Interface. 2013 Sep 6; 10(86): 20130414

(2)   The authors analyzed a very long time period covering several decades. It will be interesting if the authors could discuss the influence of demographic changes (e.g., changes in birth and death rates) on dengue transmission intensity in China.

These two references would be useful to compare: (1) Assessing the role of multiple mechanisms increasing the age of dengue cases in Thailand, PNAS (2022); (2) The impact of the demographic transition on dengue in Thailand: Insights from a statistical analysis and mathematical modeling. PLoS Medicine (2009).

Answer: Thanks for raising the issue. The limited data we collected cannot show any clear temporal change in transmission intensity in China as shown in Figure 6. It is interesting to see that “The mean age of dengue hemorrhagic fever (DHF) cases increased considerably in Thailand from 8.1 to 24.3 y between 1981 and 2017”. The data of our current study are the age-stratified serological data and cannot directly be used for analysing the age of dengue hemorrhagic fever (DHF) cases. This is surely an important question to be investigated in future. The following has been added at line 441 as a new paragraph.

“It was reported that the force of infection has declined while the average age of dengue hemorrhagic fever (DHF) cases in Thailand has increased from 8.1 to 24.3 years over last four decays. This is mainly driven by the decreased birth and death rate [69,70]. The limited data we collected in this study cannot show any clear temporal change in transmission intensity in China as shown in Figure 6.  Further our data are the age-stratified serological data and cannot directly be used for analysing the age of dengue hemorrhagic fever (DHF) cases. With the similar demographic transition due to decreased birth and death rates, it is interesting to investigate whether the similar change pattern in age of dengue cases also occurred in China. This information is important for the control and prevention of dengue in China and surely a future topic of investigation.”  

The following two papers are added in References:

  1. Cummings DAT, Iamsirithaworn S, Lessler JT, McDermott A, Prasanthong R, Nisalak A, Jarman RG, Burke DS, Gibbons RV. The Impact of the Demographic Transition on Dengue in Thailand: Insights from a Statistical Analysis and Mathematical Modeling. PLoS Med 2009; 6(9): e1000139. doi:10.1371/journal.pmed.1000139.
  2. Huang AT, Takahashi S, Salje H, Garcia-Carreras B, Anderson K, Endy T, Thomas S, Rothman AL, Klungthong C, Jones AR, et al. Assessing the role of multiple mechanisms increasing the age of dengue cases in Thailand. PNAS 2022;119(20): e2115790119. https://doi.org/10.1073/pnas.2115790119.

(3) In line 236, do you mean “credible interval”?

Answer: Yes. We thank you for your kindness to point out for us. This reminds us of the difference between the two intervals:

credible intervals incorporate problem-specific contextual information from the prior distribution whereas confidence intervals are based only on the data

All “95%CI” in the main text and appendix A have been changes into “95%CrI”.

(4) in line 99, dengue case notification data is useful to provide reliable estimation of dengue force of infection, if the model accounts for the under-reporting ratio.

Answer: We agree with this statement. When using case-notification case to estimate force of infection, the challenge is always to “account for the under-reporting ratio”. Without knowing the reliable under-reporting ratio, the estimate of force of infection using case-notification case could deviate far from the true value, see the discussion in reference [19].

Reviewer 3 Report

Where is summary statistic (mean, min, max, median, etc) of 31 data sets as completely?

Where is explanation of table 2 completely?

What is reason 9 studies, data extracted from 23 studies in China, so what is relationship?

What is 4 models related with 9 studies in geographic distribution (similarity pattern)?

Where is it (Appendix A and B), we didn't see? 

Author Response

Where is summary statistic (mean, min, max, median, etc) of 31 data sets as completely?

Answer: The original serological data of the 31 datasets are given in Appendix B. These contain the age-stratified serological data. The data divided the sample population into several age groups and gave the numbers of negative and positive samples for each age group. To estimate FOI and R0, what matters is the change of seroprevalence against age. The summary statistics are not very relevant.

Where is explanation of table 2 completely?

Answer: The explanation of Table 2 is given on lines 2989-290:

Figure 4 shows the comparison results of DIC among the four models and the final estimates (Table 2) are based on the best models that have the smallest values of DIC.

What is reason 9 studies, data extracted from 23 studies in China, so what is relationship?

Answer: At the beginning of the study, we tried getting as many relevant dengue studies as possible. At the end, we selected 23 studies that covered 9 regions (8 provinces and one region (Hong Kong)). The selection process and selection criterion are given in 2.1 Literature search and data and listed in Figure 1.

What is 4 models related with 9 studies in geographic distribution (similarity pattern)?

Answer: For the 31 data sets chosen from 23 studies, the four models, which are defined in 2.2 Dengue models, are used to fit the observational data of age-stratified serological data; the model that achieved the best performance for a data set judged by deviation information criterion (DIC) is chosen for that data set. This has been explained in 2.6 Deviation information criterion and model selection. Therefore the model used for different geographical region is selected not by geographical location but by the value of DIC.

Where is it (Appendix A and B), we didn't see? 

Answer: The appendixes A and B are provided to the journal through a package.

Round 2

Reviewer 2 Report

N/A